



# A Universal Multifractals perspective into the link between rainfall extremes and temperature

Auguste Gires[1] and Yann Torres[1,2]

[1]HM&Co, École nationale des ponts et chaussées, Institut Polytechnique de Paris, Champs-sur-Marne, France
[2]Military Institute of Engineering (IME), Rio de Janeiro, Brazil

**Correspondence:** Auguste Gires (auguste.gires@enpc.fr)

**Abstract.**

The link between rainfall extremes, usually defined as a given percentile or for a given return period, and temperature has been widely investigated using measurement data and / or convection permitting model outputs. A focus was notably on whether findings are consistent with Clausius-Clapeyron relation. A scale dependence of the rate of increase with temperature is commonly reported.

Here we investigate how rainfall extremes and more generally variability across scales change with temperature, relying on the scale invariant framework of the Universal Multifractals. Rainfall and temperature data from three high resolution measurement campaigns that took place in France between 2018 and 2025 are used. Scaling behaviour is confirmed on two distinct ranges of scales, first at event scale (30 s - 1h) and then up to synoptic scale (roughly 11 days). Then we find that across both ranges of scales, the scale invariant maximum observable singularity increases on average with greater temperature, which provides a framework to interpret previously observed trends.

## 1 Introduction

Precipitation extremes in general are expected to increase under climate change (Masson-Delmotte et al., 2021). These extremes, at sub-hourly scale, daily scale or larger scale have strong influence on river flooding, storm water management, local (including urban) flooding, debris flows, erosion... (Borga et al., 2014; Fowler et al., 2021), and can trigger in some cases natural disasters.

The main process mentioned in the literature to explain why rainfall extremes increase with temperature is thermodynamic Clausius-Clapeyron (CC) relation which quantifies the ability of warmer air to hold more moisture. It is often referred to as CC scaling, but we will not use this formulation here to avoid confusion with the scaling of rainfall fields which we will discuss later. More precisely, it states that on average, air can hold 7% more moisture per $^oC$. Actually, this rate decreases with increasing temperature. It is of 7.3 $\%^oC^{-1}$ at 0 $^oC$, 6.4 $\%^oC^{-1}$ at 15 $^oC$ and 6 $\%^oC^{-1}$ at 25 $^oC$ (Panthou et al., 2014). It is often assumed that rainfall extremes should increase at the rate suggested by CC relation. Such statements relies on three assumptions: (i) relative humidity stays roughly the same in future climate conditions, (ii) heavy rainfall events are primarily





influenced by the atmospheric water content, and (iii) atmospheric circulation patterns do not undergo significant changes in
the future climate (Panthou et al., 2014).

Numerous papers have studied the influence of temperature on rainfall extremes (usually quantified with the help of per-
centiles, typically 90, 95 or 99th; or return period) and how well CC relation is retrieved. For example, Moustakis et al. (2020)
used a combination of data and convection permitting model to show that CC expected rate holds over most part of mid and
high latitudes while deviations are found in the tropics. In a later analysis, centered on US, Moustakis et al. (2021) showed that
for a studied duration of one hour, a 20 year return rainfall event becomes a 7 year one over roughly 75% of grid points.

In an earlier work, Lenderink and van Meijgaard (2008) explained that climate models outputs seem consistent with the CC
relation, while using data from Netherlands they showed some differences according to studied time scales. For example, they
found that the increase of hourly extremes with temperature goes twice stronger than expected with CC relations. This was
further confirmed in a more recent study over Netherlands and related to the physics of convective clouds (Lenderink et al.,
2017).

Haerter et al. (2010) used 30 years of 5 min data from six stations in Germany and showed that CC relation does not provide
a good explanation at all scales. Indeed, they found a stronger increase of extremes with temperature at shorter scales and
reported continuous changes.

Chen et al. (2021) studied data from Eastern China, and also found greater increase of rainfall extremes than expected with
CC relation. Wettest 10 hours increased twice faster than CC, while the 10 heaviest daily rainfalls increase three time faster.

Changes not only with temporal scale, but also with region are found. For example Panthou et al. (2014) studied rainfall data,
and more precisely the 90 and 99th percentiles, with time steps ranging from 5 min to 12 h for more than 100 meteorological
stations across Canada. They also found that with longer durations, the increase of extreme rainfall with temperature was
less pronounced. They report differences according to regions. For example, CC relations holds for coastal regions and short
durations, while it is not the case for inland regions where super CC is observed, before an upper limit is reached.

A dependence on temperature is also observed. For example, Sharma and Mujumdar (2019) used data from India and
observed at daily scale deviations from CC relation, with stronger increase of extremes between $25^{o}C$ to $30^{o}C$, and less for
greater temperature.

Precipitations are complex as they arise from the interplay between various non-linear processes. It leads to increases or
decreases with regards to thermodynamic relation alone, i.e. CC relation. Such changes in local atmospheric dynamics explain
deviations from CC relations. As discussed above, it appears that such deviations depend on temporal scale, with shorter
duration rainfall exhibiting a increase rate stronger than expected (Fowler et al., 2021). Similar dependence on spatial scale is
also reported by Peleg et al. (2018) who studied high resolution radar data.

In order to quantify the impact of climate change on rainfall extremes, some authors used another approach. They rely
on a model for extreme value and they study the dependence of key parameters on temperature. For example, Marra et al.
(2024) fitted, on data from Switzerland, a non-asymptotic statistical model for extreme rainfall whose parameters depended on
temperature. Moustakis et al. (2021) found an increase in tail heaviness of rainfall, and related this to changes in characteristic
parameters according to temperature.





**Table 1.** Summary information for the various measurement campaigns during which the data used in this paper were collected.

| Campaign name | Start date | End day | Number of days |
|---|---|---|---|
| ENPC Campus 1 | 08/01/2018 | 22/07/2020 | 927 |
| ENPC Campus 2 | 14/10/2021 | 25/05/2025 | 1320 |
| SIRTA | 16/11/2016 | 19/09/2017 | 308 |
| Pays d'Othes | 11/12/2020 | 24/07/2023 | 956 |

In the previously mentioned studies, the underlying idea is to first properly characterize the link between rainfall extremes and surface air temperature, and then to use temperature as a proxy to predict future rainfall extremes. This study also fits in this overall context with the aim of overcoming some of the previously reported limitations.

More precisely, a key observation is that there seems to be a strong scale dependence on the increase of rainfall extremes with temperature, i.e. the increase seems stronger, and stronger than expected from CC relation only, for short duration (typically sub-hourly). Another limitations of these studies is that only a few percentiles or a few return periods are studied, and not the whole variability across scales of rainfall fields. In this paper, we suggest to investigate how rainfall extremes and more generally rainfall variability across scales, change with temperature. Indeed, rainfall is known to exhibit scale invariant features (see Lovejoy and Schertzer (1995) for an early review or Schertzer and Tchiguirinskaia (2020) for a more recent one), and relying on these features enables to suggest an innovative approach to explore the link between rainfall extremes and temperature. More precisely, this paper relies on the framework of Universal Multifractals (UM). It is a physically based, mathematically robust framework which has been designed to analyze and simulate geophysical fields exhibiting extreme variability across wide range of space-time scales such as wind (see Schertzer and Tchiguirinskaia (2020) for a review).

The paper is structured as follows. In section 2, the data from three high resolution measurement campaigns over France is presented as well selection of studied samples at synoptic and event scale. Then the methodology, is presented with a recap of basic and needed multifractal properties, and a focus on the notion of maximum observable singularity. Finally, results at both synoptic and event scales are discussed in section 4.

## 2 Data

### 2.1 Three measurement campaigns

Data collected during three measurement campaigns with devices operated as part of the Hydrology Meteorology and Complexity laboratory TARANIS observatory (exTreme and multi-scAle RAiNdrop parIS observatory) of the Fresnel Platform of École nationale des ponts et chaussées (https://hmco.enpc.fr/Page/Fresnel-Platform/en) are used. Summary information for each measurement campaign can be found in Tab. 1, and locations in Fig. 1.





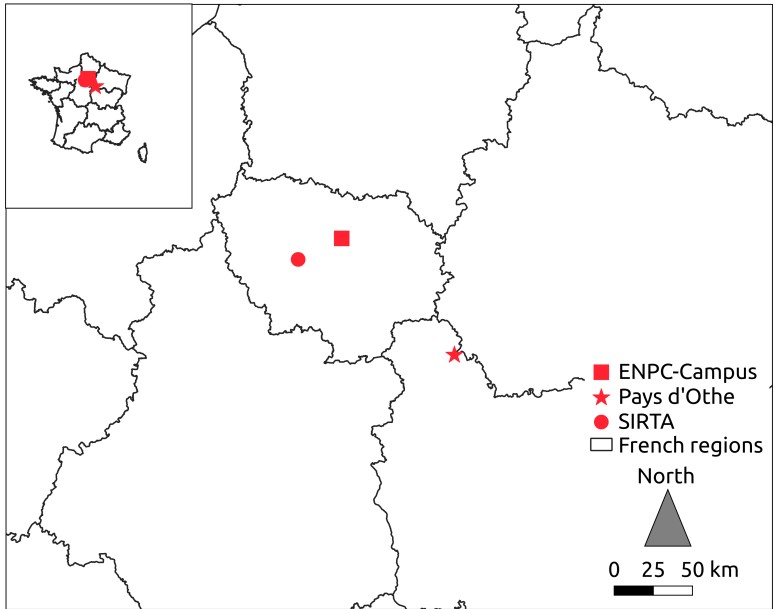

**Figure 1.** Location of the three measurement campaigns used in this paper.

The first campaign, called ENPC-Campus, takes place on the roof of the Carnot building on the campus of ENPC. In this paper, we use the rainfall data (only the rain rate in mm.h$^{-1}$) measured with the help of a Parsivel$^2$ disdrometer manufactured by OTT, and temperature obtained with the help of a sensor by Campbell Scientific. Both devices provide data with 30 s time steps. The corresponding data base, references presenting the devices, description of the campaign, as well as complete samples of data can be found in Gires et al. (2018). The rainfall and temperature time series used in this paper for the second part of ENPC-Campus campaign are displayed in Fig. 2 as an illustration.

From November 2016 to September 2017 the instruments were moved to SIRTA (Site Instrumenté de Recherche par Télédétection Atmosphérique) on the Ecole Polytechnique campus for a joint intensive measurement campaign over the Ile-de-France region, where Paris is located. The site is about 38 km away from ENPC campus towards south east of Paris. This campaign is denoted SIRTA in this paper.

The last measurement campaign used in this paper took place at a wind farm operated by Boralex and located at the Pays d'Othe (name of the campaign), approximately $120 \ km$ southeast of Paris in a slightly sloping area. As for the other campaigns, a Parsivel$^2$ disdrometer with 30 s time steps provides rainfall data. Temperature data is collected with the help of a mini meteorological station manufactured by Thies Clima and operated with a sampling rate of 1 Hz. Temperature data is upscaled to 30 s time steps to match rainfall ones. The devices were installed on a meteorological mast at a height of 45 m. Complete description of the campaign and samples of data can be found in Gires et al. (2022).

The whole rainfall and temperature times series for each measurement campaign can be accessed in Gires (2025).



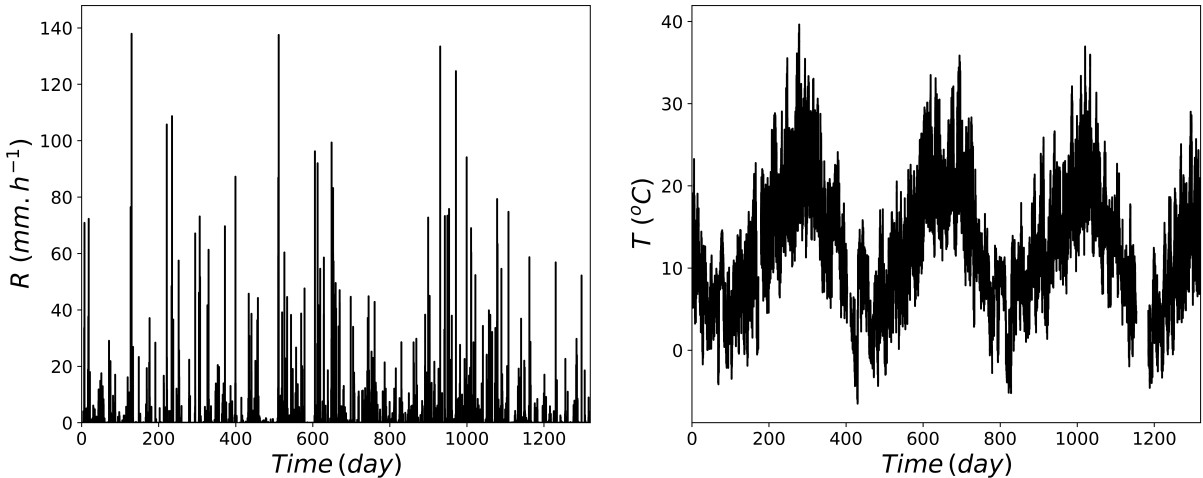

**Figure 2.** (Left) Temporal evolution of the rain rate during campaign "ENPC Campus 2". (Right) Temporal evolution of the temperature during campaign "ENPC Campus 2". 30 s time steps are used in both cases.

## 2.2 Sample selection at synoptic scale

In a first step, analysis are carried out up to synoptic scale, which corresponds to the typical duration of a meteorological situation at planetary scale. More precisely, for each measurement campaign, the whole time series is split into successive samples of $2^{15}$ time steps, which corresponds to roughly 11.4 days. The total number of samples per measurement campaign can be found in Appendix (Tab. A1).

## 2.3 Sample selection at event scale

Analysis are also implemented at event scale. To achieve this, rainfall events are selected by considering that an event is a rainy period of time during which more than 1 mm is collected and that is separated by more than 15 min of dry conditions before and after. The number of events per measurement campaign can be found in appendix (Tab. A2).

Then, a studied sample length is set. For technical reasons (see next section), it must correspond to a power of 2. Longer sample length enables to study rainfall across a wider range of scales, getting more robust results, but they impose to discard
shorter events. As a trade-off, a sample length of 128 time steps corresponding to 64 min ($\sim$ 1h) is used. Then, in order to study the maximum possible data, the process illustrated in Fig. 3 is implemented. For each event: (i) the maximum number of sub-events, i.e. non overlapping samples of size 128, is computed. (ii) The portion of length equal to the product of number of samples time 128 with highest cumulative depth is found. (iii) It is split into sample(s).





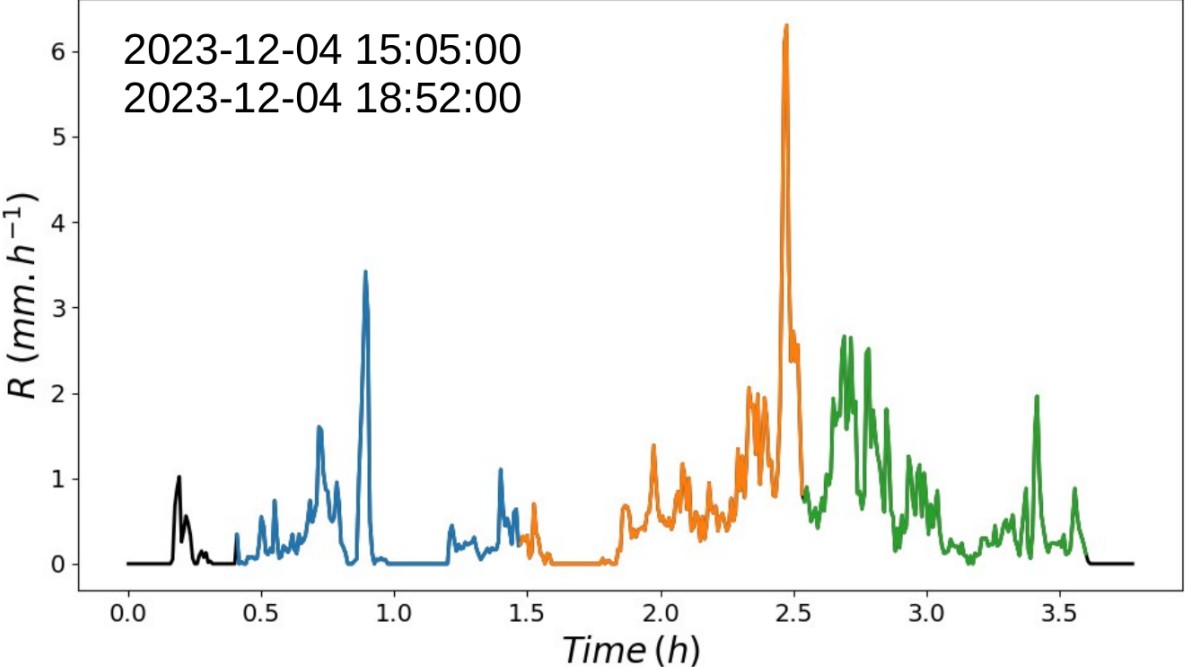

**Figure 3.** Illustration of how samples of 128 time steps (64 min) are extracted from a rainfall event that was collected during the ENPC-Campus campaign. In that case, three samples are extracted.

# 3 Methodology

## 3.1 Universal Multifractal framework

In this paper, variability and ultimately extremes of the rainfall fields across various ranges of scales, are quantified in the framework of Universal Multifractals (UM). Here, only the key elements are reminded and interested readers are referred to a recent review by Schertzer and Tchiguirinskaia (2020) and references therein for more details.

To introduce the framework, let us consider a conservative field $\epsilon_\lambda$ at a resolution $\lambda$. It is defined as the ratio between the outer scale ($T$) and observation scale ($t$), i.e. $\lambda = T/t$. For multifractal fields, the moment of order $q$ of the field is power law related to the resolution:

$$\langle \epsilon_\lambda^q \rangle \approx \lambda^{K(q)} \tag{1}$$

where $K(q)$ is the scaling moment function. It can be shown that, in an equivalent way, the probability of exceeding a scale dependent threshold ($\lambda^\gamma$) defined with the help a scale invariant singularity $\gamma$, also scales with the resolution as:



$$Pr(\epsilon_\lambda \geq \lambda^\gamma) \approx \lambda^{-c(\gamma)} \tag{2}$$

where $c(\gamma)$ is the codimension function (Schertzer and Lovejoy, 1987). The functions $K(q)$ and $c(\gamma)$ fully characterize the variability across scales of the field $\epsilon_\lambda$ and are linked by a Legendre transform (Parisi and Frish, 1985). This notably means that a singularity can be associated uniquely to each moment and vice-versa. As it can be seen on Eqs. 1 and 2, multifractal properties are statistical properties which are valid on average over numerous samples.

In the specific framework of UM (Schertzer and Lovejoy, 1987, 1997), which are a limit behaviour of all multiplicative cascades processes, $K(q)$ and $c(\gamma)$ are characterized with the help of only two parameters with physical interpretation:

- $C_1$, the mean intermittency co-dimension, which measures the clustering of the (average) intensity at smaller and smaller scales. $C_1 = 0$ for an homogeneous field;

- $\alpha$, the multifractality index ($0 \leq \alpha \leq 2$), which measures the clustering variability with regards to the intensity level.

Greater values of $\alpha$ and $C_1$ correspond to stronger extremes. For UM, we have:

$$K(q) = \frac{C_1}{\alpha - 1}(q^\alpha - q) \tag{3}$$

A Trace Moment (TM) analysis basically consists in checking the scaling behaviour of the field and estimating $K(q)$ by plotting Eq. 1 in log-log. To achieve this, the field is upscaled from its maximum resolution $\Lambda$ by averaging over adjacent time steps, then raised to various powers $q$, and finally the ensemble average (over various samples independently upscaled) is performed to obtain an estimate of the empirical moments and their scaling behaviour. UM parameters are estimated with the help of the Double Trace Moment techniques which is an extension of TM tailored for UM (Lavallée et al., 1993).

Let us now consider a non-conservative field, denoted $\phi_\lambda$, i.e. we have $\langle\phi_\lambda\rangle \neq 1$. In that case, it is usually assumed that it can be written as (with an equality in probability distribution):

$$\phi_\lambda \stackrel{d}{=} \epsilon_\lambda \lambda^{-H} \tag{4}$$

where $\epsilon_\lambda$ is a conservative field ($\langle\epsilon_\lambda\rangle = 1$) of moment scaling function $K_c(q)$ (the sub-index "c" refers to the conservativity of $\epsilon_\lambda$), and $H$ the non-conservation parameter. $K_c(q)$ only depends on UM parameters $C_1$ and $\alpha$. $H$ characterizes the scale dependence of the average field, and is equal to zero for a conservative field.

$H$ can easily be related to spectral analysis. Indeed, for scaling fields, the power spectrum follows a power law with regards to wave number:

$$E(k) \approx k^{-\beta} \tag{5}$$

and the spectral slope $\beta$ is related to $H$ with the help of the following formula (Tessier et al., 1993):





$$\beta = 1 + 2H - K_c(2) \tag{6}$$

TM and DTM technique should theoretically be implemented on a conservative field $\epsilon_\lambda$. However if $H < 0.3 - 0.4$, it can be implemented directly on $\phi_\lambda$, and will not generate biased estimates. In case of greater $H$, $\epsilon_\lambda$ should be used. Retrieving $\epsilon_\lambda$

from $\phi_\lambda$ theoretically requires a fractional integration of order $H$ (equivalent to a multiplication by $kH$ in the Fourier space). A common approximation, which provides reliable results, consists in taking $\epsilon_\Lambda$ as the absolute value of the fluctuations of $\phi_\Lambda$ at the maximum resolution and renormalizing it (Lavallée et al., 1993).

## 3.2 Maximum observable singularity

The insight one can get of a statistical process is limited by the size of the studied sample. For multifractal processes, this will

result in a maximum singularity $\gamma_s$ and corresponding moment order $q_s$ beyond which the values of the statistical estimates of respectively the codimension and scaling moment functions are not considered as reliable (Lovejoy and Schertzer, 1989, 2007).

More precisely, let's consider $N_s$ independent samples with a resolution $\lambda$. In a d-dimensional space, there are $\lambda^d$ values per sample ($d = 1$ for the time series studied in this paper). The maximum singularity ($\gamma_s$) that one can expect to observe is defined by:

$$N_s \lambda^d Pr(\epsilon_\lambda \geq \lambda^{\gamma_s}) \approx 1 \tag{7}$$

Introducing the notion of sampling dimension $d_s$: $N_s = \lambda^{d_s}$ ($d_S = 0$ for a single sample as it will be the case here), is yields:

$$c(\gamma_s) = d + d_s \tag{8}$$

which enables to estimate $\gamma_s$. For $\gamma > \gamma_s$ one expects that $c(\gamma) = +\infty$, which means that the estimates of $c(\gamma)$ will not

be reliable. As a consequence of the Legendre transform, the estimates of $K(q)$ becomes linear for $q > q_s = c'(\gamma_s)$: $K(q) = \gamma_S(q - q_s) + K(q_s)$. $\gamma_s$ quantifies the extremes that can be expected within a time series. It is especially useful to quantify how extremes evolve when $\alpha$ and $C_1$ exhibit different trends. Relying on this tools Royer et al. (2008) showed that rainfall extremes are expected to increase over France in the context of climate change. Douglas and Barros (2003) used it to discuss the concept of maximum probable rainfall. Qiu et al. (2024) relied on this tool to quantify the impact of rainfall space-time variability on

the usefulness of natured-based solutions in urban environment.





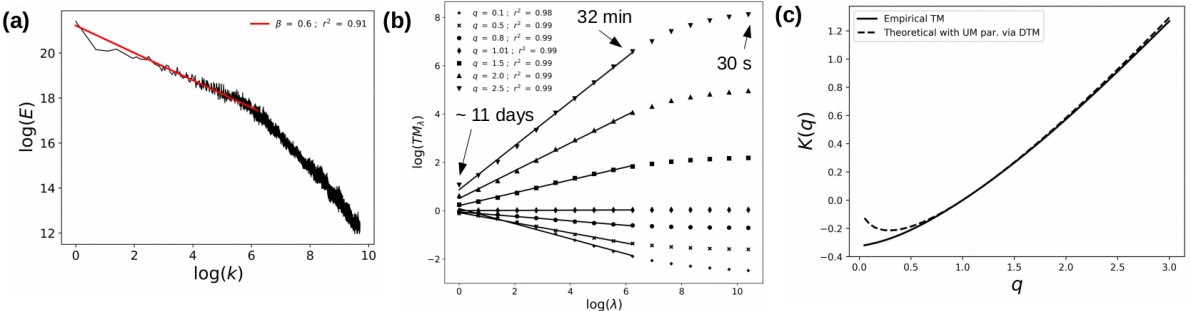

**Figure 4.** For the campaign "ENPC campus" with ensemble analysis at synoptic scale. (a) Spectral analysis, i.e. Eq. 5 in log-log. (b) TM analysis, i.e. Eq. 1 in log-log. (c) Scaling moment function $K(q)$: empirical estimate and theoretically fitted shape using UM parameters from DTM analysis.

## 4 Results

### 4.1 Synoptic scale

In this subsection, we implement multifractal analysis up to synoptic scale (see section 2.2). In a first step, an ensemble analysis is carried out, that is to say all samples are upscaled independently and used to compute average statistical moments in Eq. 1 or spectra in Eq. 5. Such analysis is used to study general features of scaling.

Let us illustrate the results with the ENPC-Campus campaign. Outcome of spectral analysis, i.e. Eq. 5 in log-log, is displayed in Fig. 4.a. An excellent scaling behaviour on scales ranging from roughly 1/2 h to 11 days is found, and smaller scales will be investigated in next subsection. The spectral slope $\beta$ is smaller than 1, meaning that on this range of scales the studied field is conservative and multifractal analysis can be implemented directly on the field. Trace Moment analysis (i.e. Eq. 1 in log-log) outcome is displayed in 4.b. The coefficients of determination of the linear regressions are all greater than 0.99 for $q > 0.5$ and we use the one for $q = 1.5$ as a metric. Results confirm the excellent scaling behaviour on this range of scale.

UM parameters estimated with the help of DTM technique are reported in Tab. 2. These values are typical for this range of scales (Ladoy et al., 1993; de Lima and Grasman, 1999). An excellent agreement is retrieved when comparing the empirical scaling moment function $K(q)$ derived from TM analysis and the theoretical one plotted using Eq. 3 and DTM estimates of UM parameters (Fig. 4.c). The discrepancies that can be noticed for $q < 0.5$ are explained by a multifractal phase transition associated with the numerous zeros which are in the time series (see Gires et al. (2012) for more details on this phenomenon). Very small values of $H$ (last column of Tab. 2), i.e. smaller than 0.1 corresponding to almost conservative field, are retrieved on this range of scales.

Same excellent scaling behaviour is observed on this range of scales for the two other measurement campaigns. Similar UM parameters are retrieved with only limited variations of $\alpha$ and $C_1$ (see Tab. 2).




**Table 2.** Summary of UM parameters assessed at synoptic scales (32 min - 11 days) using ensemble analysis

| Campaign name | $r^2$ | $\alpha$ | $C_1$ | $\beta$ | $H$ | $\gamma_s$ |
|---|---|---|---|---|---|---|
| ENPC Campus | 0.997 | 0.746 | 0.448 | 0.582 | 0.076 | 2.93 |
| SIRTA | 0.997 | 0.639 | 0.488 | 0.549 | 0.0732 | 3.07 |
| Pays d'Othe | 0.997 | 0.746 | 0.448 | 0.608 | 0.0956 | 3.19 |

**Table 3.** Slope ($\times 10^{-2 \circ}C^{-1}$) (corresponding $R^2$) of the linear regression of the value of the studied parameter vs. $<T>$ (individual sample analysis) at synoptic scale. Illustration in Fig. 5 for "ENPC" Campus campaign.

| Campaign name | $\alpha$ | $C_1$ | $\beta$ | $H$ | $\gamma_s$ |
|---|---|---|---|---|---|
| ENPC Campus | -1.29 (0.131) | 1.52 (0.281) | -1.90 (0.0943) | -0.209 (0.00782) | 1.07 (0.302) |
| SIRTA | -0.0700 (0.000) | 0.478 (0.0364) | -2.226 (0.195) | -0.841 (0.201) | 0.516 (0.0927) |
| Pays d'Othe | -1.28 (0.113) | 1.39 (0.221) | -1.93 (0.129) | -0.302 (0.0233) | 0.947 (0.222) |
| All | -1.21 (0.112) | 1.41 (0.240) | -1.94 (0.108) | -0.283 (0.0162) | 0.994 (0.259) |

In a second step, a UM analysis is implemented on each sample individually using the same range of scales. In addition the average temperature $<T>$ for each sample is computed from the available data. Individual samples with bad scaling, i.e. with $r^2 < 0.9$ for $q = 1.5$ are discarded (see appendix for numbers of samples kept in analysis). Scatter plots of retrieved UM parameters vs. $<T>$ are displayed in Fig. 5 for ENPC-Campus campaign. Significant scattering is observed for all parameters. Potential overall trends are identified with the help of a simple linear regression. It is displayed through the red line on the plots. Assessed slope and corresponding coefficient of determination $r^2$ are displayed in Tab. 3.

The $r^2$ coefficients are low, which is expected given the observed scattering. It means that the retrieved trends are only valid on average over numerous events. We observe a decreasing trend for $\alpha$ and an increasing trend for $C_1$. Hence the consequences on extremes are not obvious and $\gamma_s$ is needed to combine the effects of both. It appears that $\gamma_s$ exhibits an increasing trend with $<T>$. It means that stronger variability and extremes are retrieved on studied samples with increasing temperature. A slightly decreasing trend is found for $\beta$ and very slightly decreasing one for $H$.

## 4.2 Event scale

In this subsection, analysis are carried out at event scale (see section 2.3). The same analysis as for longer samples at synoptic scale are carried out, with ensemble analysis first, to identify scaling behaviour, and then individual sample analysis.

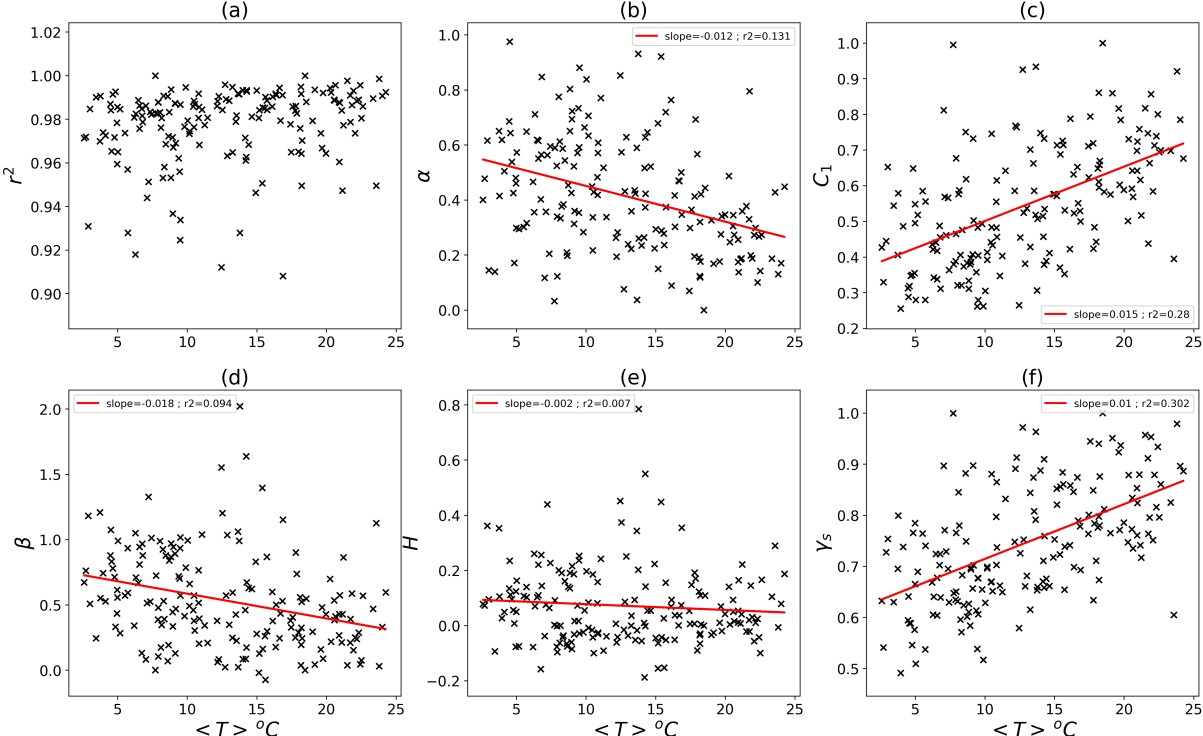

**Figure 5.** For the campaign "ENPC campus" with individual sample analysis at synoptic scale: $r2$ for $q = 1.5$ in TM analysis (a), $\alpha$ (b), $C_1$ (c), $\beta$ (d), $H$ (e) and $\gamma_s$ (f) vs. $< T >$

Results are illustrated with the data from ENPC-Campus campaign. Spectral analysis (Fig. 6.a) shows that data exhibits a very good scaling behavior on the whole range of scales from 30 s to 1 h. The spectral slope $\beta$ is of 1.73 meaning that on this range of scales, the studied field is not conservative. Hence the analysis is done on the conservative part (see section 3.1). Fig. 6.b shows TM analysis. The excellent scaling behaviour on the whole range of studied scales (30 s - 64 min) is confirmed, with coefficients of determination all greater than 0.99 for $q > 0.5$.

UM parameters obtained via DTM analysis are in Tab. 4. These values around 1.7-1.8 for $\alpha$ and 0.2 for $C_1$ are consistent with those commonly reported in the literature for this range of scales (Gires et al., 2016; de Montera et al., 2009; Mandapaka et al., 2009; Verrier et al., 2010; Jose et al., 2024). As for the results at synoptic scale, there is a very good agreement between the empirical scaling moment function $K(q)$ and the theoretical one. Some differences become visible for $q \approx 2.3 - 2.5$ which is slightly smaller than the expected value of $q_s$ equal to 2.7 in this case.

Very similar qualitative and quantitative (see Tab. 4) are retrieved on this range of scales for the two other measurement campaigns.

As for the synoptic scale, a UM analysis is implemented on each event individually using the same range of scales and the average temperature $< T >$ for each event is also assessed. Individual samples with bad scaling, i.e. with $r^2 < 0.9$ for $q = 1.5$,





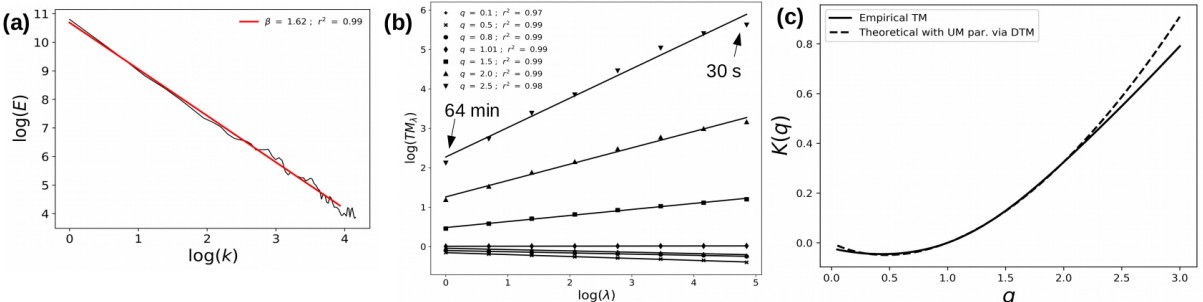

**Figure 6.** For the campaign "ENPC campus" with ensemble analysis at event scale. (a) Spectral analysis, i.e. Eq. 5 in log-log. (b) TM analysis, i.e. Eq. 1 in log-log. (c) Scaling moment function $K(q)$: empirical estimate and theoretically fitted shape using UM parameters from DTM analysis.

**Table 4.** Summary of UM parameters assessed at event scale (30 s - 64 min) using ensemble analysis

| Campaign name | $r^2$ | $\alpha$ | $C_1$ | $\beta$ | $H$ | $\gamma_s$ |
|---|---|---|---|---|---|---|
| ENPC Campus | 0.994 | 1.73 | 0.180 | 1.63 | 0.478 | 2.70 |
| SIRTA | 0.988 | 1.75 | 0.190 | 1.50 | 0.421 | 2.59 |
| Pays d'Othe | 0.995 | 1.88 | 0.206 | 1.52 | 0.459 | 2.31 |

and / or average temperature lower than 2 $^o C$ are discarded (see appendix for numbers of samples kept in analysis). The latter
condition enables to focus only on rainfall events and avoid looking at snowfall ones.

Scatter plots of retrieved UM parameters vs. $< T >$ are displayed in Fig. 7 for ENPC-Campus campaign. Stronger scattering
than for the synoptic scales is retrieved. Similarly potential overall trends are computed with the help of linear regressions.
Slopes and $R^2$ are reported in Tab. 5 for all measurement campaigns.

In general similar results but with less pronounced trends (smaller $r^2$) are retrieved for UM parameters with a slightly
decreasing $\alpha$, an increasing $C_1$ and an increasing $\gamma_s$. It means that extremes and variability also tend to increase with tem-
perature over this range of scales. Contrarily to what is observed at synoptic scale, $\beta$ and $H$ are increasing with temperature,
corresponding to a greater non conservativeness of the field.

### 4.3 Link with other studies

As discussed in the introduction, numerous studies report a scale dependence of the increase of rainfall extremes with temper-
235 ature, i.e. that the increase is stronger in percentage for shorter durations (Fowler et al., 2021; Haerter et al., 2010; Lenderink
and van Meijgaard, 2008; Lenderink et al., 2017; Panthou et al., 2014).





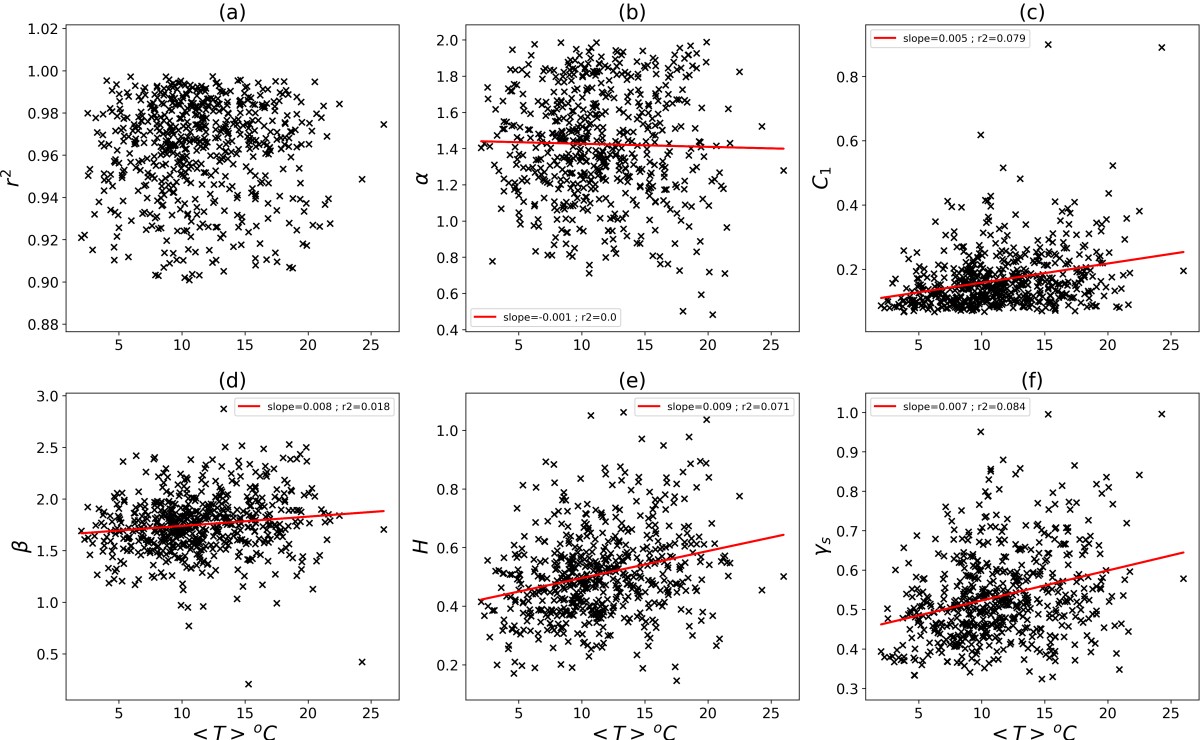

**Figure 7.** For the campaign "ENPC campus" with individual sample analysis at event scale: $r^2$ for $q = 1.5$ in TM analysis (a), $\alpha$ (b), $C_1$ (c), $\beta$ (d), $H$ (e) and $\gamma_s$ (f) vs. $< T >$

**Table 5.** Slope ($\times 10^{-2 \circ} C^{-1}$) (corresponding $r^2$) of the linear regression of the value of the studied parameter vs. $< T >$ (individual event analysis) at event scale. Illustration in Fig. 5 for "ENPC" Campus campaign.

| Campaign name | $\alpha$ | $C_1$ | $\beta$ | $H$ | $\gamma_s$ |
|---|---|---|---|---|---|
| ENPC Campus | -0.170 (0.001) | 0.597 (0.0799) | 0.897 (0.0181) | 0.922 (0.0714) | 0.761 (0.0843) |
| SIRTA | -1.39 (0.0414) | 0.587 (0.162) | 1.60 (0.0944) | 1.20 (0.214) | 0.664 (0.102) |
| Pays d'Othe | -1.15 (0.0277) | 0.346 (0.0293) | 1.82 (0.063) | 1.11 (0.0956) | 0.285 (0.0114) |
| All | -0.592 (0.00728) | 0.508 (0.0626) | 1.31 (0.0371) | 1.03 (0.089) | 0.589 (0.0518) |

In this study we find in general an increase of the scale invariant concept of maximum observable singularity $\gamma_s$ of rainfall time series with temperature. The rainfall extreme that one can expect to observe in a sample at resolution $\lambda$ behaves as $\lambda^{\gamma_s}$. We remind that $\lambda$ is the resolution, i.e. the ratio between the outer scale and the observation scale, and that it increases with shorter duration. Hence, an increase of the scale invariant $\gamma_s$ with temperature results in greater increase of extreme rainfall in





percentage at higher resolution, i.e. with shorter observation scales. Indeed, this percentage of increase $\%_{incr}$ can be written as:

$$\%_{incr} = 100 \times \left( \lambda^{\gamma_s(T_1) - \gamma_s(T_2)} - 1 \right) \tag{9}$$

for a change from temperature $T_1$ to $T_2$. Hence the changes with temperature in the scale invariant $\gamma_s$ provide a framework
to explain changes in increase of rainfall extremes with temperature according to scale (mainly from hourly to daily) which are reported in previous studies.

## 5   Conclusions

In this paper, we study how rainfall extremes and more generally variability across scales changes with temperature. For this, we use data coming from three high resolution measurement campaign that took place in France between 2018 and 2025; and
we rely on the framework of Universal Multifractals. More precisely, we first confirm scaling behaviour and then estimate UM parameters $\alpha$, $C_1$, the corresponding maximum observable singularity $\gamma_s$, and $H$ for each sample and study their dependence to temperature.

It appears that for scales ranging from 32 min up to the synoptic scale of roughly 11 days, an excellent scaling behaviour is retrieved and we observe a decrease of $\alpha$ with average temperature, an increase of $C_1$ which yields an increase of $\gamma_s$. There
is a slight decrease of $H$. Similar trends but less pronounced are observed at event scale, i.e. for scales ranging from 30 s to roughly 1 h, for $\alpha$, $C_1$ and $\gamma_s$. On the contrary, an increasing trend with average temperature is found for $H$.

This increase of $\gamma_s$ with temperature confirms previous findings of expected increase of rainfall extremes with temperature. It confirms them in a scale invariant way and enables to explain the dependence of the rate of increase with observation scale that is reported in previous studies.
Consistent results are found here between event and synoptic scales and over three measurement campaigns. It suggests that findings are robust. It would be relevant to expand the analysis to much wider areas using data from various climates to expand our understanding on the dependence of rainfall extremes with temperature. Investigating the geographical dependence of the rate of change of UM parameters with temperature would notably be insightful and should be pursued in upcoming studies.

*Code and data availability.*  Data used in the paper, i.e. the rainfall and temperature time series with 30 s time steps for the three measurement
campaign, along with a python scrip containing the functions needed to implement the spectral and multifractal analysis carried out in this paper can be found in Gires (2025).



**Table A1.** Number of studied samples in individual sample analysis at synoptic scale.

| Campaign name | # of sample in total | # of events after removing too short ones or missing $T$ | # events after criteria on $r^2$ and $< T >$ |
|---|---|---|---|
| ENPC Campus | 197 | 196 | 172 |
| SIRTA | 27 | 27 | 20 |
| Pays d'Othe | 84 | 84 | 84 |

**Table A2.** Number of studied event in individual event analysis at event scale.

| Campaign name | # of event in total | # of events after removing too short ones or missing $T$ | # events after criteria on $r^2$ and $< T >$ |
|---|---|---|---|
| ENPC Campus | 785 (2071) | 644 | 583 |
| SIRTA | 113 (207) | 84 | 74 |
| Pays d'Othe | 389 (1043) | 316 | 288 |

## Appendix A: Appendix A

*Author contributions.* A.G. designed the initial content of the study. Y.T. implemented the initial version of the study on RW-Turb campaign under supervision of A.G.. A.G. extended it to the other campaigns and wrote the paper. Y.T. reviewed the paper.

*Competing interests.* Authors declare they have no competing interests

*Acknowledgements.* The authors acknowledge partial financial support from the Chair of Hydrology for Resilient Cities (endowed by Veolia) of the École nationale des ponts et chaussées, EU NEW INTERREG IV RainGain Project, EU Climate KIC Blue Green Dream project, the Île-de-France region RadX@IdF Project, and the ANR JCJC RW-Turb project (ANR-19-CE05-0022-01); which enabled the collection of data. Authors acknowledge the France-Taiwan Ra2DW project, supported by the French National Research Agency (ANR-23-CE01-0019-275 01) for partial financial support.



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
