# Peer review of "A Universal Multifractals perspective into the link between rainfall extremes and temperature"

_EGUsphere, 2025_

## Author Comment (AC1)

We would like to thank you for your review. Please find below our answers to your comments. The manuscript was updated accordingly.

**General comments**

The reviewed manuscript aims at investigating how rainfall extremes (more generally variability across scales) vary with temperature utilizing the scale invariant framework of the Universal Multifractals. This topic is of interest to a wide audience of hydrometeorologists. The Authors use rainfall and temperature data from three high resolution measurement campaigns that were conducted in France covering the period from 2018 to 2025. Their findings suggest that: a) scaling behavior is confirmed on two distinct ranges of temporal scales (event scale and synoptic scale) and b) the invariant maximum observable singularity is increasing on average with respect to temperature. The manuscript is well structured and and clearly organized, so no changes are suggested.

Thank you for your positive comments !

**Specific comments**

Identification of potential trends of UM parameters with respect to temperature $T$ is conducted by applying linear regression (see Lines 199 – 201 and 226 – 228 and also Figures 5 and 7). In all cases considered, significant scattering is observed, associated with low values of the coefficient of determination. It is my opinion that it would be helpful if a statistical test was applied to quantify the level of significance of rejection of the null hypothesis of no trend.

Following your suggestion, a statistical test was applied (p-value<0.05) was used to reject the null hypothesis of no trends. This is now clarified in the text. In addition, a new analysis based on samples based on bins of temperature was carried out and presented.

**Technical corrections**

Line 13 and other places were few references are cited: Add more references or use "see e.g."

This was done, thank you for your careful reading.

Line 15: Periods after the word erosion should be replaced by etc.

This was done, thank you for your careful reading.

Line 39: Replace "greater" by "more pronounced"

This was done, thank you for your careful reading.

Line 55: Remove "they"

This was done, thank you for your careful reading.

In Figure 2 replace "temporal evolution" with "time series"

This was done, thank you for your careful reading.

---

## Author Comment (AC2)

We would like to thank you for your review. Please find below our answers to your comments. The manuscript was updated accordingly.

This manuscript investigates the relationships between the parameters of Universal Multifractal (UM) models applied to rainfall time series and temperature measurements. These UM models are applied to rainfall observations at a very fine temporal resolution over the period 2018-2025, and recorded at three different locations near Paris, France. The manuscript is well structured, and the authors have provided a summarised description of UM which is not too technical and focuses on the interpretations of the results (parameters, scaling properties). I must acknowledge that I am not an expert in UM. It seems that the description of the relationships between the UM properties and temperature has never been done before, and I imagine that this study is of interest for readers who are well versed with this type of model. However, in my opinion, the authors miss the opportunity to reach a more general audience because these results are difficult to interpret in comparison to other approaches. My recommendation is thus to extend the end of the study. What is missing is an interpretation of the UM results in terms of rainfall intensity (return levels), at different durations (see major comment #1).

Thank you for your positive comments on the manuscript.
With regard to the second part, see answer to MC#1.

In addition, I did not understand if these results can be related to the Clausius-Clapeyron relationship. There is a very short subsection dedicated to this question (subsection 4.3) but no results are provided (e.g. application of Eq. 9). Additional results illustrating how UM properties can be related to the Clausius-Clapeyron relationship, i.e. if they confirm the CC scaling discussed at l. 18-21 (see major comment #2) are needed.

See answer to MC#2.

**MC#1:** The manuscript put a very strong emphasis on rainfall extremes (title, abstract, introduction, etc.). The abstract starts with a definition of rainfall extremes: "a given percentile or for a given return period" so that rainfall extremes must be interpreted as large intensities with a rare frequency. Typically, return periods are expressed in years (e.g. intensities occurring once every 10 years on average). At l. 59-61, it is indicated that the aim of the study is to properly characterize the link between rainfall extremes and temperature. However, the rainfall series are rather short (6 years, <1 year and 2.5 years) and the rest of the study does not provide results on large return levels. The beginning of the manuscript is thus misleading in that respect. As I understand, the UM perspective considers that the properties of rainfall extremes can be derived from the scaling behaviour of rainfall from a very fine temporal resolution (30 s) to aggregated time steps. What is very difficult is to understand what the results shown in Figs. 5 and 7 mean in broader terms, for readers unfamiliar with UM, e.g. in terms of rainfall intensities at an hourly or daily time scale, and for return periods of 2, 5 or 10 years. I also mention this aspect because the trends for alpha and C1 are not easy to interpret in both Figures (opposite trends in Fig. 5, weak trends in Fig. 7) and the trend for gamma_s is very weak in Fig. 7 (r2=0.084). At the end of the section "Results", it is indicated that this study provides "a framework to explain changes in increase of rainfall extremes with temperature according to scale (mainly from hourly to daily) which are reported in previous studies". In my opinion, additional results must be shown to demonstrate that this coherence exists, e.g. the rate of increase of rainfall extremes per degree, for a given return period, at an hourly and daily time scales, for the three locations, derived from the UM, and in comparison to standard approaches using longer time series (precipitation and temperature) nearby. I certainly understand that this asks for additional work but in my opinion, the current manuscript is too much focused on

the application of the UM framework and does not show the general interest of this framework for studying the link between rainfall extremes and temperature.

With regard to rainfall extremes and IDF curves. The paper aims at exploring rainfall extreme variability through UM framework. The latter can be used to characterize IDF curves, but this is outside the scope of the paper. A paragraph was added in the conclusion to clarify this as a relevant perspective (authors fully agree that it is indeed a topic to be fully explored in a dedicated paper!): "Practitioners are typically used to work with IDF curves and how they evolve with climate change. In order to further link the results obtained in this paper with previous findings, often expressed in terms of return periods, it would be needed to continue exploring the theoretical links between UM parameters and IDF curves. This is a topic currently under investigation, relying on initial elements to be further developed from \citet{bendjoudi_interpretation_1997} or \citet{Langousis_2009}. Once this is achieved, the trends in UM parameters could be simply input in the theoretical shape of IDF curves and results quantitatively compared with existing work. This corresponds to stimulating work to be carried out."
The title was also updated to avoid any confusion. The abstract clearly states the scope of the paper "Here we investigate how rainfall extremes and more generally variability across scales change with temperature". A few sentence in the introduction were also updated to avoid any confusion. With regard to the robustness of trends. Following your comment and some of another reviewer, a additional analysis relying on samples binned by temperature intervals was done and added to the paper (new Fig. 6).

**MC#2:** There is almost no discussion of the results in broader terms at the end of the paper. In particular, the CC relation is discussed in depth in the introduction and I was expecting an interpretation of the results in terms of CC relation at the end of the paper: is the UM framework in coherence with the CC relation or does it lead to a super-CC, a sub-CC? I also miss a general discussion about the impact of climate change on the relation between rainfall extremes and temperature for different types of rainfall events (https://doi.org/10.1007/s40641-015-0009-3).

The analysis carried out do not aim at determining whether a super-CC or sub-CC is observed. Following your comment, this was clarified in the beginning of section 4.4 (formerly 4.3). With regard to rainfall type, some analysis focusing only on highest rainfall was carried and did not yield different results. Splitting the temperature range was also explore, as well as seasonal effect. A new subsection was added to present the results. Thanks for the suggestion of citation that was incorporated in this new subsection.

**Minor comments:**
l. 2: "convection permitting model outputs" -> or more generally climate model outputs: https://doi.org/10.1029/2019GL082908.

The sentence was updated in the abstract.
Thanks for the suggestion of paper. It was added.

l. 52: "a" -> an

Thanks for your careful reading ! This was corrected.

l. 128: I suggest replacing "as it can be seen on Eqs. 1 and 2" by "Eqs. 1 and 2 also mean that" because I do not think many readers will see that.

Thanks for your suggestion which we implemented.

l. Eq. 3: missing dot at the end of the equation.

This was updated.

l. 153: is it H<0.3 or H<0.4? Or maybe rephrase as H is roughly smaller than [0.3, 0.4]?

This was updated following your suggestion.

l. 182: /2 h -> 30 minutes.

This was updated following your suggestion.

Table 3: Why is the slope for alpha so different for the SIRTA campaign? It looks like the linear model failed (r2 for alpha equals to zero), could you comment on this in the manuscript?

A filtering for potential snowfall was also added in the larger scale analysis and values of alpha and C1 and now very similar for the three campaigns.

---

## Author Comment (AC3)

We would like to thank you for your review. Please find below our answers to your comments. The manuscript was updated accordingly.

**Summary:** The paper studies the relationship between rainfall extremes and temperature using the Universal Multifractals (UM) framework. Using high-resolution rainfall and temperature data from 3 campaigns in France, the authors confirm robust scaling behavior at both the event scale (30s to 1h) and the synoptic scale (up to ~11 days). They further show that the maximum observable singularity tends to increase with temperature. The study argues that UM-based analyses provide a convenient, scale-invariant approach to understand the temperature dependence of rainfall extremes, in a way that is consistent with earlier findings based on Clausius–Clapeyron scaling.

**Critical assessment:** The application of the UM formalism to study the link between rainfall extremes and temperature is relatively new. The observation that the maximum observable singularity may be temperature-dependent is potentially interesting, though similar ideas have already been presented in other UM/rainfall studies in the broader climate context. The paper does not advance the UM methodology itself; it simply applies an existing framework to a new dataset and context. The study is strongly limited by its geographical scope (only 3 sites in France) and by the modest length of the underlying time series. Moreover, the practical implications of the findings are not obvious to me, and the authors do not really articulate them well in the paper. The paper is full of typos and difficult to read. Several parts of the methodology are poorly written and hard to understand, even for specialists (see comments below). The introduction provides a good summary of prior CC-based studies, but overall I get the feeling that the authors overstate the relevance of their results. As far as I understood it, the UM framework does not really provide any new insights into the rainfall-temperature link. Please correct me if I am wrong. If the results are just consistent with what is already known, what's the added value from a scientific point of view? What are the pros and cons of this framework, and what issues/questions remain open?

Results offer a new insight into existing ones by focusing on the extreme variability which was not previously addressed. This was clarified (see also detailed answer below). Perspectives were added in the conclusion.

**Major comments:**
**(MC1):** The paper contains many typographical mistakes and grammatical errors. A thorough proofreading and, ideally, professional language editing would be highly beneficial to improve clarity and readability. See Typos for some examples.

The manuscript underwent a thorough proofreading.

**(MC 2):** Instrumental/observational uncertainty is not quantified or discussed. Please provide rough estimates of the uncertainties affecting your data and discuss how this might affect modeled quantities (see MC 3).

Following your comment a paragraph was added in the data section mentioning recent references addressing the issue of instrumental limitations with disdrometers. Addressing these issues and the influence of the associated uncertainty on the multifractal analysis carried out would be an interesting topic, but it remains outside the scope of this paper.  It is now clarified in the same paragraph.

**(MC 3):** No error bars, confidence intervals, or uncertainty estimates of estimated UM parameters ($\alpha$, C1, H, $\gamma$s) are presented. This makes it difficult to judge the significance of the observed trends. Please provide rough estimates of uncertainties and how they might affect your conclusions.

Following your comments, rough estimates on the uncertainty were computed for UM parameters alpha and C1. They are discussed on the ensemble analysis at both event and large scale. There are very limited with regard to discussed trends so they do not affect our conclusions. This is now clarified in section 4.1 and 4.2.

**(MC 4):** The description of the different quality control mechanisms and methods for filtering our bad data (e.g., due to solid precipitation, small sample sizes, negative temperatures etc..) needs to be extended. In the Data (or Methods) section, please provide a clear step-by-step description of all the filters that were applied. Currently, the information is scattered across the Results.

Following your comment, the filtering / preparation of the data (temperature <2°C, event selected, sample selection…) was clarified and all moved to data section. Only the comments on samples / events with poor scaling were kept in the results section because they correspond to results.

**(MC 5):** The paper would benefit from a more thorough discussion about the limitations of the proposed approach. For instance, the assumption of stationarity (seasonal and diurnal variability) and the treatment of mixed precipitation types (rain vs. snow, briefly mentioned on page 11, line 224) could be addressed in greater detail and with a more critical perspective. Similarly, the strong reliance on surface temperatures without any consideration for vertical variability constitutes another limitation. Not all precipitation extremes are generated by the same physical mechanisms, and not all events at a given temperature are comparable from a physical point of view. The paper should clearly acknowledge and discuss the critical assumptions underlying such an analysis.

With regard to the treatment of mixed precipitation types, only rainfall events are considered and this was clarified in the data section. A comment on the use of only surface temperature was added in the introduction.

**(MC 6):** The paper would benefit from a short, additional analysis of scaling rates of the 95% and 99% quantiles of rain rates with temperature at a few key time scales. This analysis could be added to Section 4.3 (Link with other studies) or presented at the start of the Results section, to provide more context and better understand how the new findings from the UM framework complement traditional CC-scaling analyses.

Following your comment, a subsection was added to present results according to seasons (mentioned in MC5) and rainfall intensities. Given the limited number of events available, only the 90% quantiles were used for rainfall intensity and this is explained in the text.

**(MC 7):** The reference list contains numerous formatting inconsistencies (e.g., journal names, DOIs, URLs) and requires careful revision (see technical comments at the end of the review). A thorough check against the journal's style guide would improve consistency and readability.

This was corrected.

**Minor, technical comments:**
- The terminology in Section 3 is confusing. The authors use "fields" to refer to time series. Yes, the theory of UMs is applicable to any type of stochastic process (including spatial processes), but this

paper only deals with time series. Therefore, "time series" or "stochastic process" would be more appropriate.

Fields can be 1D, 2D or more in general; hence we prefer to keep the standard terminology in UM literature. Yet, you are correct that only time series are studied in this paper. Therefore, following your comment, we added a sentence: "The word "field" can refer in general to processes in 1D, 2D or more; yet in this paper, only 1D processes corresponding to time series are analysed." in section 3.1 to avoid any confusion.

- Page 2, line 28: The assumption that extreme precipitation rates should increase at the same rate than predicted by CC also relies on the assumption that surface temperatures are a good indicator of total precipitable water in a column of air. This may not be the case for all types of rainfall extremes, especially at daily and longer time scales where atmospheric dynamics and large-scale circulations play a much more important role than temperature. Please reformulate the text accordingly.

The corresponding paragraph was updated to account for your comment.

- Page 2, ll. 30-34: in the study by Lenderink and Meijgaard (2008), it is important to mention that the 2CC scaling only holds for a particular temperature range, and only for the higher quantiles during the warm season.

This was clarified.

- Page 2, line 36: you could mention the reply by Haerter & Berg (2009) to the paper by Lenderink and Meijgaard (2008), in which they labeled the 2CC scaling a "statistical" artifact. Haerter & Berg argue that 2CC scaling is not physical. It arises from the superposition of two different rainfall regimes (stratiform and convective), both of which exhibit CC scaling on their own, albeit with different magnitudes. Because the ratio of stratiform to convective rain gradually decreases with increasing temperature, the net scaling rate can reach 2 CC over a limited range of temperatures. Reference: Haerter, J., Berg, P. Unexpected rise in extreme precipitation caused by a shift in rain type? Nature Geosci 2, 372-373 (2009), https://doi.org/10.1038/ngeo523

Thank you for pointing this paper which we had missed. It was added in the corresponding paragraph to strengthen the discussion.

- In the introduction, the scale break at higher temperatures (e.g., decrease in scaling rate of precipitation extremes beyond 26°C potentially leading to zero or negative scaling) should be mentioned and discussed in more depth. There are plenty of studies that have looked at the sensitivity of scaling rates to the choice of the temperature range. Please pick a few and include them into your literature review. In Europe, temperatures above 25°C are often associated with high-pressure systems, which inhibit convection. At higher temperatures, other crucial factors such as the dew point temperature and atmospheric stability are therefore needed to understand the relationship between peak precipitation rates and temperature. The highest temperatures you consider during the event analysis seems to be around 25°C. It would be interesting to know what happens beyond that.

Thanks for the suggestion. We added a paragraph in the introduction. No clear effect of potential temperature regimes was found and this is now mentioned in the new subsection.

- Page 5, line 100: "In a first step, analysis analyses are carried out up to synoptic scale, which corresponds to the typical duration of a meteorological situation at planetary scale".
This sentence is very confusing. Synoptic scale usually refers to phenomena that last 2-7 days (cyclones, fronts etc..) and extend over spatial scales of 100-1000 km. Planetary scale refers to phenomena that last for weeks to months and extend over much larger spatial scales (jet stream, planetary waves etc..). Please reformulate to clarify what you meant.

This was clarified following your comment. For better clarity, this regime is now denoted "large scale" in the paper with an upper limit corresponding to typical synoptic scale.

- Page 5, ll. 105-107: The definition of a "rain event" could benefit from further clarification. As it stands, events could overlap in time, with starting times potentially differing by only a single time step. However, since Table A2 reports only a few hundred events, it appears that some procedure may have been used to avoid overlapping events. Please clarify the procedure you used for identifying and selecting events."

In order to avoid any confusion, this was clarified following your comment.

- Page 5, ll. 111-112: Please specify how the starting times of these samples were determined. Diurnal variations in rainfall intensity/variability may impact the results depending on how the data were resampled.

The process is initiated at the beginning of the available data without accounting for potential effect of diurnal variations. This was clarified in section 2.2 of the updated manuscript.

- Page 5, ll. 111-113: The procedure for selecting sub-events with a fixed length of 128 samples needs further clarification. The phrasing is awkward and the illustration in Figure 3 does not really help understand how the method works. If I understood correctly, for each rain event, you try to partition the event into as many non-overlapping samples of length 128 as possible. You then look for the partition that contains the heaviest rainfall chunk. Please clarify to avoid any misunderstandings! Also, please explain what to do in case two or more partitions have the same max rainfall value.

Yes, you are correct. The paragraph was updated to improve clarity following your comment.

- Page 7, line 142: The notion of a conservative versus non-conservative fields should be explained earlier. Also, the meaning of the operator <> should be explained. Since many readers may not be familiar with this notation, I suggest to use the standard, expected value operator instead.

We believe that it is easier for the reader to introduce first the UM framework on conservative fields and address in a second stage the topic of conservative vs. non-conservative fields. This topic can be updated if needed.
Following your suggestion the meaning of <> was explained at its first use in Eq. 1. We prefer to keep this notation, which is commonly used in UM literature.

- Page 8, line 154: "[…] and will not generate biased estimates". This statement is too strong, as some bias will be introduced. I suggest to write: "without substantially biasing the estimates of alpha and C1."

Indeed you are correct. This was updated following your suggestion.

- On Zenodo, please zip the csv data files. This reduces file sizes by at least an order of magnitude and will make it much easier for people to download and store the data.

Thank for your suggestion, this will be done at a later stage once the final version of the paper is available.

- Please pay more attention to verb tenses. In the Introduction, some present tense statements are mixed into a past-tense literature review. My recommendation: use past tense for literature review (except general truths). The methods section mostly uses present tenses, but a few sentences slip into past tense. Please use present tenses wherever possible. In the Results, you inconsistently use past tenses ("was found") and present tenses ("is found"). My recommendation is to use past tenses for findings, and present tenses for figure/table descriptions.

This was done, following your suggestions.

- Fig1: some information is missing on the map, such as geographical coordinates and/or names of departments/regions.

Following your suggestion, geographical coordinates were added as well as region names.

- A histogram of surface temperatures during rainy periods, as well as a scatterplot of rain rate distribution versus temperature for the events mentioned in table A1 and A2 would be useful, to get a sense of the temperature range over which precipitation occurred.

We are not sure to understand your suggestion, because the temperatures are already visible in Fig. 5 and 7. The scatterplot you have in mind would be for each time step or each event / sample (in that case should the cumulative rainfall or average or maximum rainfall be plotted ?) ?

**Typos and grammatical mistakes:**
Please be aware that this is not an exhaustive list!
**Page 1**
- Line 22: "Such statements relies" → **"Such statements rely"**
**Page 2**
- Line 33: "goes twice stronger" → **"is twice as strong as"**
- Line 34: "over Netherlands" → **"over the Netherlands"**
- Line 40: "Wettest 10 hours" → **"The wettest 10 hours"**
- Line 45: "CC relations holds" → **"CC relation holds"**
- Line 49: "Precipitations are complex" → **"Precipitation patterns are complex"**
- Line 52: "a increase rate" → **"an increase rate"**
**Page 3**
- Table 1: "End day" → **"End date"**
**Page 4**
- Line 95: "times series" → **"time series"**
- Line 96: "to match rainfall ones" → **"to match the resolution of the rainfall data."**
**Page 5**
- Line 105: "Analysis are also" → **"analyses are also"**
**Page 6**
- Line 120: "is power law" → **"is a power law"**

**Page 7**
- Line 150: "with regards to" → **"with regard to"**
**Page 8**
- Line 166: "is yields" → **"it yields"**
- Line 174: "this tools" → **"this tool"**
- Line 176: "natured-based solutions" → **"nature-based solutions"**
**Page 10**
- Line 210: "behavior" → **"behaviour"** (if keeping UK spelling)
**Page 12**
- Line 237: "tem-perature" → **"temperature"** (hyphenation error)
**Page 14**
- Line 253: "changes" → **"change"** (singular subject earlier in sentence)
**Page 15**
- Line 265: "campaign" → **"campaigns"** (plural, since referring to three campaigns).
**Appendix**
- Table A1 & A2: "# of sample" → **"# of samples"**, "# of event" → **"# of events"**.

Thank you for your careful reading, this was updated.

**References:**
- Several references have strange DOIs starting with <GotoISI>. Please check that the URLs are correct.

Thanks for your careful reading, this was corrected.

- Journal names are inconsistently formatted: some are full names (Journal of Hydrology) while others are abbreviated (J. Hydrometeorol., Wat. Resour. Res.). Please use consistent formatting and names.
- Several references contain URLs of the form "http://www.sciencedirect.com/…". These should be replaced with the actual DOI of the article, as specified on the publisher's webpage: For example: "https://doi.org/10.1016/j.advwatres.2012.03.026" instead of "http://www.sciencedirect.com/science/article/pii/S0309170812000814".

Thanks for your careful reading, this was corrected.

- **Borga et al. (2014)** includes "climatic change impact on water: Overcoming data and science gaps" at the end, which is weird. Maybe some leftover text?

Thanks for your careful reading, this was corrected.

- **Douglas & Barros (2003) has d**uplicated journal name entries: Journal of Hydrometeorology, 4, 1012–1024, j. Hydrometeorol., 2003.

Thanks for your careful reading, this was corrected.

- **Haerter et al. (2010) the** DOI has redundant parts https://doi.org/https://doi.org/…

Thanks for your careful reading, this was corrected

**- Masson-Delmotte et al. (2021, IPCC) has a double comma in** "Yu, R., , and Zhou, B."

Thanks for your careful reading, this was corrected

**- Moustakis et al. (2021)** includes repeated identifiers (e2020EF001824 2020EF001824).

Thanks for your careful reading, this was corrected

**- Panthou et al. (2014)** Title ends with an asterisk (*). Not sure why.

Thanks for your careful reading, this was corrected

**- Parisi & Frish (1985) I believe that** "Frish" is misspelled. Should be "**Frisch"**

Thanks for your careful reading, this was corrected

**- Sharma & Mujumdar (2019) the DOI has redundant parts** https://doi.org/https://doi.org/

Thanks for your careful reading, this was corrected

---

## Author Comment (AC4)

We would like to thank you for your review. Please find below our answers to your comments. The manuscript was updated accordingly.

The manuscript aims to study the link between precipitation extremes and temperature with a scale invariant framework based on the Universal Multifractal. The analysis is carried out on 3 high resolution time series of precipitation available in the Paris region, France.
I am not an expert of it but the theoretical UM framework seems to be well established and to allow for such an analysis. I see however some limitations in the analysis that should be likely fixed or at least discussed to strengthen the potential impact of this work and make it suitable for publication in HESS.

Thank you for your suggestions to strengthen the discussion part of this paper. See our answers below.

The innovation with respect to previous works is not clear for me. It has to be clarified in the introduction. What results (findings, robustness of findings, multiscale coherency of results ?) are allowed by this UM based analysis that could not have been presented in other previous works – especially with respect to the Temp/PrecipExtremes relationship.

In this paper, we suggest to address the same issues as previously mentioned authors and to not focus on a few single observation scales independently as usually done, but to investigate how rainfall extremes and more generally rainfall variability across scales, change with temperature. This will enable to get more robust results in the sense that they are valid across a given range of scales.
Introduction was updated to better clarify this.

The dependency to temperature is explored here with observations. Observations are obviously key for this. Other high resolution time series of precipitation are available worldwide. The work would really gain value and generality if other stations, from other climate contexts could be integrated in the analysis. The 3 stations considered here belong to a same and very small meteorological region and one would likely have some comparative results / findings of analyses for other contexts.
For me also, the authors should also recognize / discuss the interest of climate simulations for this T/Pextreme exploration (with strengths / limitations compared to analyses based on observations), especially those produced from convection permitted models. Works based on climate model outputs are numerous to quantify the impact of climate change on rainfall extremes. Models come of course obviously with a number of limitations but they give the opportunity to explore a much larger "meteorological/climatic domain" than those available from observations. Using models may also allow to explore the importance of the limitations mentioned in the introduction ln 23-24 (especially those relative to possible change in circulation regimes). I would strongly suggest to include a discussion on those issues, at least to mention them.

Yes you are right that the studied time series are short and come from a small geographical area. We agree that it would be interesting to carry out the analysis on other time series. The use of climate model outputs would also be very interesting, we agree. Following your suggestion, the conclusion was updated to include in more details these suggestions as interesting future work.

To estimate possible evolutions of extreme with climate change, an alternative to climate models is indeed that of the statistical approaches mentioned ln 55, where precipitation characteristics are regressed against temperature. As statistical approaches, they obviously present also a number of limitations that should be acknowledged. The most critical one is likely the assumption that the

 identified from observations will be still valid in a modified climate (stationarity assumption). This assumption will likely not hold in a number of regions, especially (but not only) as a result of changes in circulation regimes. Different evolutions of this relationship may likely exist depending on how circulation regimes will change (this is likely to be shown with climate experiments with different climate models). This issue should be likely commented, at least mentioned in the introduction or elsewhere.

Indeed the assumption of stationarity is a strong limitation of the approach. Following your comment (which was also pointed out by another reviewer), this is now mentioned in the introduction.

For the introduction and perhaps discussion, I would thus suggest to put in a larger perspective the issue targeted in the manuscript, clearly identifying what we know / do not know for the present / future climates, what data / tools we have to explore this issue, what are the knots and challenges for scientists there, etc…

The paragraph introducing the work was updated to improve clarity.

In the introduction, the authors have a long discussion on the Clausius-Clapeyron relationship and on previous analyses of its interest to support observations of changes in precipitation extremes worldwide. How results of the present work confirm / contradict this CC influenced behavior of extremes for the considered stations ? Is there any dependency on time resolution ? on season, weather/rainfall type ? on the spatial integration area of real interest for extremes (local extremes are of little interest for most "impacts"). Those points would be worth a section in the discussion. The discussion should be likely strengthened. It would be worth to better put in perspective the results of the work with those of other studies / other approaches. Are the identified trends similar / smaller / larger than in the other works ? What is the added value of the present work ?  The discussion should also discuss the limitations of the work

Results confirm the influence of temperature on rainfall extremes, as expected from CC relationship. The scale invariant framework used enables to have this result obtained for various time resolution. This is clarified in the text (section 4.4).
With regard to season effect, a section has been added to do discuss this following your comment (as well as one from another reviewer).
For quantitative comparison with existing work, a full theoretical shape of IDF curves with UM parameters would be needed. This is an ongoing work. Following your comment, a paragraph was added in the conclusion as perspective.

The perspectives of the work would be also worth to be mentioned in the conclusion.

Following your comment, conclusion was significantly expanded to include more discussion on potential perspective of this work.

Detailed comments.
Equation 9 : the % increase is valid for all return periods ? how does it compare to other works where the % increase has been sometimes found to depend on T (e.g. Chagnaud et al. 2025)

No dependency was found on temperature range in this study. This issue is now discussed in the new section 4.3, which was added following your remarks.

Ln 51 – 54 : the rationale / scope of the paragraph is not clear. A reformulation would be worth.

We are sorry, but we are not sure to understand what you mean.

Ln. 59. How is it possible to "properly" characterize the link ? Are we sure that the link exists ? that it is strong ? I guess this is not always the case. What about the significance of the link ?

Following your comment, the sentence was rephrased and qualified to avoid any confusion.

Ln 64. "Another limitation" : what is the first one ?

It was corrected to "A limitation"

Ln 102 and 107. The number of years considered should be given in the main text.

This information was added.

Ln 112-113. Can you clarify "point ii)" ? I did not understand what is done / why this is done.

This was clarified.

Fig. 3. I do not understand why a sequence of 4 hours of rain should be split into 3 different subevents. Is it relevant to consider that the temperature predictor can be considered with a so small resolution (i.e. that changes in temperature from one 3h time step to the other can have some explanatory potential on precipitation extremeness ?) Can you clarify ?

This is done to have a unique scale range for all the event to have consistent analysis on the UM analysis. In event analysis, the samples of a same event are analysed together and the average temperature considered is the one for the whole event. This was clarified in section 4.2.

Ln 119. What is a conservative field and ln 142 : what is a non conservative field ??? For me, rainfall is by nature conservative. (at least observations). Can you define "conservative" ?

Following your comment, the paragraph was updated, and an additional reference containing details for interested reader cited.

Ln 212 and 231 I do not understand how a field can be non-conservative. Please clarify.

It corresponds to fields with long range correlation. It was clarified in the text.

Ln 128. How is defined a "singularity"

A "singularity" can be seen as a scale invariant threshold. More precisely, a multifractal fields behaves as lambda^singularity. The singularity remains the same across scales while the value of the field changes with scales; which is typically the case for rainfall. 20 mm/h does not have the same meaning over a time step of 30 s and 1 day. Following your comment, this was clarified in the text.

Ln 130. I am not sure I agree. Is it valid for all multiplicative random cascades processes ? microcanonical ones ? canonical ones ?

UM are limit behaviour of all multiplicative random cascade processes. It comes from a generalization of the central limit theorem. The precise meaning is described in details in the reference from 1997 which is already cited. This was clarified in the text with a clear reference for the interested reader.

Ln 130. I am not sure multiplicative random cascades have been defined previously.

Indeed, but we believe that it is not needed to add more details for this paragraph which remains very generic. This can of course be updated if you believe it is much needed.

Ln 154. "in case of greater H, epsilon should be used". Should be used to do what ?

To implement TM and DTM technique. This was clarified.

Ln 177. "synoptic scale" : warning, this is only the temporal scale. All your analysis are on local data (then no synoptic in spatial dimension)

Yes you are correct. Following your comment and also one by another reviewer, this was clarified in the data section. More generally, the term "large scales" for the regime 30 min – 11 days is now used to avoid any confusions.

Table 2. There is some large difference in the coefficients between stations (while the 3 stations belong to a same very small "climate" region). Can you comment ? Is it expected ? large ? small ? reasons for this ?

Values were recomputed after adding a filtering on snowfall also for long samples and they are now very similar.

Ln 197 and 223 : "individual sample with "bad" scaling". I do not understand why you expect "good" scaling behavior for all events. The variability of rainfall temporal patterns is potentially huge and if I understand that a scaling behavior is expected in average considering all events, I guess there is no reason to have it for each event. Then, how do you deal with those events which do no have a scaling behavior ??? I fear that disregarding them may lead to too naïve interpretations of the existence / strength of the temperature / scaling relationship. Can you clarify, discuss this point.

Scaling being an average behaviour, it is expected that some samples will not exhibit a good scaling behaviour. And they were incorporated in the ensemble analysis. Yet, adding their assessed parameters in the individual analysis would not be relevant because they are not reliable so they could bias results. This was clarified in the text.
Yet, following your suggestion, an additional type of analysis was carried out, enabling to account for all the events. It consists in performing a UM analysis on ensembles of sample / event binned within temperature intervals. In enables to get more robust results. So thank you for your stimulating question !

Ln 202-206. Can you comment more divergent results on C1 and alpha ? Could you have some equifinality issue here ? can you precise why taux-s is a better variable to study the relationship ?

Divergent results on alpha and C1 are common, and already reported in Royer et al. (2008) for example. It is not an equifinality issue. gamma_s combines the influence of both which is why it is interesting to use.  Section 3.2 was updated to clarify this.

Ln 231 : what is the significance of the estimated trends ?

A statistical test to assess the significance of the retrieved trends was added in the paper and is now discussed.

Chagnaud et al. 2025. How fast is the frequency of daily rainfall extremes doubling in global land regions, *ERC*. https://doi.org/10.1088/2515-7620/ad9f12

Thank you for your suggestion. This paper is now discussed in the text.